# Expression of STING in Women with Morbid Obesity and Nonalcoholic Fatty Liver Disease

**DOI:** 10.3390/metabo13040496

**Published:** 2023-03-29

**Authors:** Laia Bertran, Laia Adalid, Mercè Vilaró-Blay, Andrea Barrientos-Riosalido, Carmen Aguilar, Salomé Martínez, Fàtima Sabench, Daniel del Castillo, José Antonio Porras, Ajla Alibalic, Cristóbal Richart, Teresa Auguet

**Affiliations:** 1Grup de Recerca GEMMAIR (AGAUR)—Medicina Aplicada (URV), Departament de Medicina i Cirurgia, Institut d’Investigació Sanitària Pere Virgili (IISPV), Universitat Rovira i Virgili (URV), 43007 Tarragona, Spain; 2Servei Anatomia Patològica, Hospital Universitari Joan XXIII Tarragona, Mallafré Guasch, 4, 43007 Tarragona, Spain; 3Servei de Cirurgia i Anestèsia, Hospital Sant Joan de Reus, Departament de Medicina i Cirurgia, Universitat Rovira i Virgili (URV), IISPV, Avinguda Doctor Josep Laporte, 2, 43204 Reus, Spain; 4Servei de Medicina Interna, Hospital Universitari Joan XXIII Tarragona, Mallafré Guash, 4, 43007 Tarragona, Spain

**Keywords:** nonalcoholic fatty liver disease, stimulator of interferon genes, lipogenesis, inflammation, microbiota

## Abstract

Nonalcoholic fatty liver disease (NAFLD) is the most prevalent chronic hepatic disease. Although mostly benign, this disease can evolve into nonalcoholic steatohepatitis (NASH). The stimulator of interferon genes (STING) plays an important role in the immune response against stressed cells, but this protein may also be involved in liver lipogenesis and microbiota composition. In this study, the role of STING in NAFLD was evaluated by RT–qPCR to analyze STING mRNA abundance and by immunohistochemical analysis to evaluate protein expression in liver biopsies from a cohort composed of 69 women with morbid obesity classified according to their liver involvement (normal liver, n = 27; simple steatosis (SS), n = 26; NASH, n = 16). The results showed that STING mRNA expression in the liver increases with the occurrence of NAFLD, specifically in the SS stage in which the degree of steatosis is mild or moderate. Protein analysis corroborated these results. Positive correlations were observed among hepatic STING mRNA abundance and gamma-glutamyl transferase and alkaline phosphatase levels, hepatic Toll-like receptor 9 expression and some circulating microbiota-derived bile acids. In conclusion, STING may be involved in the outcome and progression of NAFLD and may be related to hepatic lipid metabolism. However, further studies are needed to confirm these findings.

## 1. Introduction

Nonalcoholic fatty liver disease (NAFLD) has been reported to be one of the most common liver diseases worldwide [1]. Its prevalence is constantly increasing due to the adoption of nonhealthy lifestyles, including a nonequilibrated diet and sedentarism [2], which have become more common in developed countries [3,4,5]. NAFLD is caused by an excessive accumulation of fat in more than 5% of hepatocytes, which indicates simple steatosis (SS) [6]. In addition, NAFLD entails the absence of a significant daily consumption of alcohol or the presence of different causes of secondary hepatic steatosis [4,7]. NAFLD might progress to the nonalcoholic steatohepatitis (NASH) stage, which is also characterized by the occurrence of cellular stress, inflammation, hepatocyte ballooning, and, sometimes, fibrosis [8,9,10]. If NASH development is not prevented, this condition may progress to cirrhosis and finally to hepatocellular carcinoma (HCC) [11,12].

Microbiota may play an important role in NAFLD development [13,14]. Poor alimentary habits can induce a change in the microbiome termed intestinal dysbiosis [15]. The permeability of the intestinal barrier increases as a consequence of microbiota dysregulation. This phenomenon leads to the release of some microbiota-derived mediators, such as bile acids, short-chain fatty acids, bacterial DNA, and lipopolysaccharides, into the blood circulatory system [16,17], after which they can enter the liver through the portal vein [13,18]. Once in the liver, hepatic Toll-like receptors (TLRs) can detect these mediators and trigger the production of proinflammatory cytokines that initiate hepatic steatosis and inflammatory processes that are characteristic of NASH [19].

Unfortunately, although NAFLD is highly prevalent, no specific pharmacological treatment for NAFLD has been accepted by regulatory agencies, and thus, it is necessary to perform more extensive studies to find potential therapeutic targets [20]. In this sense, the stimulator of interferon genes (STING), also known as transmembrane protein 173 (TMEM173), is a protein strongly involved in host defense mechanisms against pathogens and against surrounding stressed cells and stressed mitochondria [21,22]. This protein has been implicated in the innate immune system [23], where it functions to boost the production of type I interferon (IFN) [24]. The process by which STING functions starts with the detection of external DNA or excreted mediators from stressed cells through the cGAS pathway [25], which triggers the downstream activation of STING. This signaling cascade is responsible for the release of type I IFN and proinflammatory factors through intermediates of the pathway including IFN regulatory factor (IRF3) and TANK binding kinase 1 (TBK1) [26,27,28].

NAFLD may be related to innate immune-mediated sterile inflammation [29], and IFNs have been shown to play an essential role in the progression of NAFLD [30]. Additionally, cGAS-STING signaling might be involved in the development of NAFLD through DNA-mediated type I IFN production [31]. Hepatic STING expression has been demonstrated to be upregulated in patients with NAFLD [32]. Moreover, the STING-IRF3 axis is involved in the activation of apoptotic pathways in NAFLD, where it also upregulates inflammatory cascades and induces glucose and lipid metabolism disorders [22].

Proinflammatory factors, such as tumor necrosis factor-α (TNF-α) and interleukin (IL)-1β and IL-6, are released through STING signaling and are involved in NAFLD progression [33]. However, STING seems to present a distinct purpose in different stages of NAFLD. On the one hand, STING is related to lipid metabolism and is implicated in the process of lipid storage since it interacts with enzymes involved in lipogenesis [25,34]. On the other hand, a greater proinflammatory engagement of STING is observed in the early stages of inflammation and fibrosis in the NASH stage [35,36,37]. Nonetheless, STING seems to play a protective role in the last stage of the disease, HCC, by behaving as an antitumoral agent [35]. In addition, STING has also been suggested to influence gut microbiota modulation [38], and this microbiome dysregulation may be strongly associated with NAFLD pathogenesis [13].

Considering the different roles that STING could play in the outcome and progression of NAFLD, the aim of the current work is to provide new perspectives regarding the hepatic mRNA and protein expression of STING in a cohort of women presenting with morbid obesity (MO) and different NAFLD stages.

## 2. Materials and Methods

### 2.1. Subjects

This research was evaluated by the institutional review board (Institut Investigació Sanitària Pere Virgili (IISPV) CEIm; 23c/2015; 11 May 2015). All participants provided their written consent after being properly informed. A total of 69 Caucasian women with MO (BMI > 40 kg/m^2^) constituted the cohort. Through a planned laparoscopic bariatric surgery, liver biopsies were obtained, always when liver disease is suspected and required for a clinical diagnosis. Exclusion requirements were: (1) patients with neoplastic, infectious, or acute or chronic hepatic diseases; (2) consumption of more than 10 g/day of ethanol or other toxins; (3) patients receiving fibrates; (4) diabetic women receiving insulin or another medication that can modify endogenous insulin levels, (5) women who use contraceptives or menopausal women.

### 2.2. Sample Size

GRANMO calculator has been employed to evaluate the sample size. An alpha risk of 0.05 and a beta risk of less than 0.2 in a bilateral contrast has been accepted. The sample size needed to detect a statistically significant difference between two proportions is 20 subjects in the control group (NL) and 40 subjects in the study group (NAFLD), with a proportion of 0.31 and 0.69 between control and study groups. In this analysis, ARCSINUS approach has been used.

### 2.3. Liver Pathology

Hematoxylin and eosin and Masson’s trichrome stains were used to classify liver samples as previously reported [39,40], and afterwards, they were assessed by an experienced hepatopathologist. Women with MO were categorized into NL (control group) (n = 27) and NAFLD (n = 42) based on their hepatic histology. NAFLD patients were subdivided into two subgroups: SS (micro/macrovesicular steatosis without fibrosis or inflammation, n = 26) and NASH (Brunt grades 1–2, n = 16). No porto-portal fibrosis was present in any of the patients with NASH employed in this work.

### 2.4. Biochemical Analyses

From the entire study population, physical, anthropometric, and biochemical evaluations were conducted. Prior to bariatric surgery, specialized nurses took blood samples using a BD Vacutainer^®^ device after the patients had fasted for the previous night. Venous blood samples were collected in tubes with or without ethylenediaminetetraacetic acid, and they were then centrifuged (3500 rpm, 4 °C, 15 min) to separate the plasma and serum aliquots. Biochemical parameters were analyzed using a conventional automated analyzer. A homeostatic model assessment for IR was applied to estimate IR (HOMA1-IR). Following the manufacturer’s instructions, cytokines including IL-1, IL-6, IL-8, IL-10, IL-17, TNF-α and adiponectin were measured using multiplex sandwich immunoassays and the Millipore Human Adipokine Magnetic Bead Panel 1 (HADK1MAG-61K), Millipore Human High-Sensitivity T Cell Panel (HSTCMAG28SK), and the Bio-Plex 200 instrument Liquid chromatography coupled to triple-quadrupole-mass spectrometry was used to analyze the absolute quantification of circulating bile acids such as glycochenodeoxycholic acid (GCDCA), glycocholic acid (GCA) and glycodeoxycholic Acid (GDCA), taurocholic acid (TCA), tauroursodeoxycholic acid (TUDCA), glycoursodeoxycholic acid (GUDCA), choline, trimethylamine (TMA), trimethylamine N-oxide (TMAO), betaine, and SCFAs (LC-QqQ). The Centre for Omic Sciences assessed each of these analyses (Rovira i Virgili University-Eurecat).

### 2.5. Gene Expression in Liver

Hepatic samples from the patients were gathered during bariatric surgery and processed and stored at −80 °C using liquid nitrogen. The total RNA was extracted from liver through using the RNeasy mini kit (Qiagen, Barcelona, Spain). The High-Capacity RNA-to-cDNA Kit was employed for reverse transcription to obtain cDNA (Applied Biosystems, Madrid, Spain). The detection of STING (Hs00736955_g1) in liver was performed by a real-time quantitative PCR (RTqPCR) by using the TaqMan Assay, which was predesigned by Applied Biosystems. Assessment of other genes involved in hepatic lipid metabolism was carried out, such as sterol-regulatory-element-binding protein 1c (SREBP1c) (Hs01088691_m1), liver X receptor alpha (LXRα) (Hs00173195_m1), and fatty acid synthase (FAS) (Hs00188012_m1). Other related hepatic genes, such as toll-like receptor (TLR) 2 (TLR2) (Hs02621280_s1), TLR4 (Hs00152939_m1), and TLR9 (Hs00370913_s1), were assessed as well.

The cycle threshold (Ct) values were recorded, normalized to housekeeping gene expression 18S RNA (Fn04646250_s1), and transformed to relative gene expression value using the 2^−ΔΔCt^ method. Then the expression of each gene was normalized using the control group (NL) as a reference. In 96-well plates, all the reactions were duplicated by applying the QuantStudio™ 7 Pro Real-Time PCR System (Applied Biosystem, Foster City, CA, USA).

### 2.6. Immunohistochemistry Analysis

To determine the protein expression of STING, an immunohistochemical analysis (IHC) was carried out in hepatic tissue classified according to the histology of the liver in NL (n = 6), SS (n = 6) and NASH (n = 6). First, the samples were fixed in 10% formaldehyde. Liver sections were stained for IHC using the Ventana Benchmark ULTRA autostainer. Primary antibody STING/TMEM173 (HPA038116), host: rabbit, supplied by Sigma-Aldrich, was diluted 1:50 for IHC using Ventana Antibody Diluent.

Anti-TMEM173/STING positive cells were evaluated under a microscope (×400) by a specialist hepatopathologist. The immunoreactive index was evaluated based on the degree of STING cytoplasmic positivity in nonparenchymal liver cells.

### 2.7. Statistical Analysis

The statistical program SPSS/PC+ for Windows was used to analyze the data (version 27.0; SPSS, Armonk, NY, USA: IBM Crop). The Kolmogorov–Smirnov test was performed to determine the distribution of the variables. The median and interquartile range were utilized to express all results (25th–75th). Besides, differences between groups were evaluated by the Mann–Whitney U test. The Spearman’s method was applied to determine the strength of the correlation between the variables. Statistical significance was defined as *p*-values < 0.05. GraphPad Prism software was employed to plot the graphics (version 7.0; GraphPad, San Diego, CA, USA).

## 3. Results

### 3.1. Baseline Features of Subjects

The biochemical and clinical parameters were determined in the cohort of women with MO (body mass index, BMI > 40 kg/m^2^) who were categorized as having normal liver (NL, n = 27), SS (n = 26) or NASH (n = 16) based on hepatic histology, using hematoxylin and eosin and Masson’s trichrome stains (Figure 1).

Nonsignificant differences were found in body weight, BMI, systolic and diastolic blood pressures (SBP and DBP), insulin, glycosylated hemoglobin (HbA1c), cholesterol, high-density lipoprotein cholesterol (HDL-C), low-density lipoprotein cholesterol (LDL-C), aspartate aminotransferase (AST), alanine aminotransferase (ALT), and gamma-glutamyl transferase (GGT). Table 1 shows that higher levels of glucose and alkaline phosphatase (ALP) were found in SS subjects than in NL patients; moreover, higher levels of triglycerides (TG) were found in NASH patients than in NL patients. Additionally, it was revealed that ALP levels were higher in NASH patients than in those with SS.

### 3.2. Evaluation of the Relative mRNA Abundance of STING in Relation to Hepatic Histology

To fulfil the purpose of this research, that is, to examine the role of STING in NAFLD, the mRNA expression of STING was determined in liver tissues in a cohort of women with MO.

First, the hepatic expression of STING in patients with MO and NL was compared with that of patients with MO and NAFLD. A significant increase in the relative mRNA expression of STING was found in NAFLD patients compared with NL subjects (Figure 2A).

To further explore this finding, the cohort was subdivided based on hepatic histopathology (NL, SS, and NASH). A significant increase in STING mRNA expression was observed in SS subjects compared with NL subjects, and a significant decrease in mRNA abundance was observed in subjects with NASH compared with subjects with SS (Figure 2B). Despite this finding, the differences in STING expression between the NL and NASH cohorts were not significant.

### 3.3. Evaluation of the Relative mRNA Abundance of STING in Relation to the Severity of Steatosis

Since the relative expression of STING was increased at the SS stage of NAFLD in hepatic tissue from women with MO, the steatosis parameter was used to categorize patients into four groups according to severity: absence, mild, moderate, and severe. A significant increase in STING mRNA expression was found in subjects with a mild degree of steatosis compared with subjects without steatosis. Moreover, another significant increase was observed in patients with moderate stages of steatosis compared with those without steatosis. No other significant differences between steatosis groups were identified (Figure 3).

### 3.4. Evaluation of the Relative mRNA Abundance of STING in Relation to NASH-Related Parameters

Previous studies have reported a possible association between STING function and NASH processes [35,36,37]. Accordingly, the relative mRNA abundance of STING was assessed in accordance with NASH-related factors, including lobular and portal inflammation and hepatocellular ballooning. On the one hand, when examining relative STING mRNA expression, the mRNA levels were not significantly different between liver samples with inflammation and those without inflammation. Nonsignificant differences were identified when the inflammation was categorized as either lobular and portal. On the other hand, no significant difference was observed in STING expression in the liver samples between the groups with regard to the absence or presence of hepatocellular ballooning. None of our patients presented with liver fibrosis.

### 3.5. Correlation of Relative mRNA Abundance of Hepatic STING with Clinical and Biochemical NAFLD-Related Features

Since it has been reported that STING can induce proinflammatory mediator release [35,36,37] and is related to lipid metabolism [25,34] and changes in the intestinal microbiota [38], all of which are mechanisms involved in NAFLD progression, correlations between STING and inflammatory cytokines, liver enzymes, lipid metabolism-related genes, and hepatic Toll-like receptors and some microbiota-derived metabolites, such as bile acids, were determined. Positive correlations were identified between the relative mRNA expression of hepatic STING and the levels of liver enzymes such as GGT and ALP (Figure 4A,B). Additionally, a positive correlation between hepatic STING and hepatic TLR9 expression was observed (Figure 4C). Moreover, hepatic STING expression was shown to be positively correlated with circulating levels of bile acids such as GCDCA, GCA, GDCA, TCA, TUDCA, and GUDCA (Figure 4D–I). However, we could not demonstrate any relationship between hepatic expression of STING and the primary genes involved in hepatic lipid metabolism (SREBP1c, *p* = 0.498; LXRα, *p* = 0.884; and FAS, *p* = 0.137).

### 3.6. Assessment of STING Protein Expression According to Liver Histology through the IHC Analysis in Liver Samples

In order to complete our results on the relationship between STING and the development of NAFLD, we studied the protein expression of STING by IHC analysis. STING protein expression was examined in NL, SS, and NASH liver sections (Figure 5A–C). Liver sections with SS and NASH revealed increased STING protein expression in nonparenchymal liver cells (mainly immune cells: macrophages/Kupffer cells and endothelial cells), compared to the NL group. Specifically, the SS group stands out, with a much higher protein expression than in the other groups.

## 4. Discussion

In this study, in a well-characterized cohort of women with MO who were at different NAFLD histological stages, an increase in hepatic STING mRNA and protein expression was found, especially in the steatosis stage. The novelty of this work is that we report the correlation between STING expression and hepatic TLR9 expression and the levels of several circulating microbiota-derived bile acids.

Given that STING has been tightly linked to the secretion of proinflammatory factors and the immune response [21], to find new therapeutic targets, the potential role of STING in NAFLD pathogenesis was evaluated in this study. The results obtained showed that relative hepatic STING mRNA expression was increased in the presence of NAFLD, specifically in the SS stage, which was reinforced with the protein expression analysis. A study by Luo et al. reported that STING expression was higher in liver tissue samples from patients with NAFLD than in those from patients without NAFLD, which is in agreement with our findings. In the same study, they used mouse models to show that the main features of the disease, such as plasma levels of ALT, hepatic steatosis and inflammation, are ameliorated when STING expression is inhibited [32]. Hence, in liver tissue, STING appears to be associated with outcomes of NAFLD.

Subsequently, increased STING mRNA and protein expression was observed in the SS stage when the relationship between STING abundance and different degrees of NAFLD was evaluated. This result appears concordant with the finding that STING plays a key role in lipid metabolism [25,34]. In particular, some evidence indicates that STING strongly interacts with acetyl-CoA carboxylase (ACC) and fatty acid synthase (FASN), both of which are enzymes involved in lipid synthesis, which suggests a key role of STING in lipid metabolism [34].

Nevertheless, this work reported a decrease in hepatic STING expression in NASH compared with SS. This result is controversial, since Wang et al. reported a higher number of liver STING-positive cells in NASH patients who presented with inflammation and liver fibrosis [41]. However, the cohort enrolled in the current study did not present with liver fibrosis, and this could probably be one of the reasons of this contradiction.

The abovementioned results suggest that STING in this precise cohort of women with MO seems to be mainly involved in lipid accumulation rather than in inflammatory processes. Similarly, Vila et al. reported that STING interferes with lipid homeostasis in the absence of proinflammatory stimuli, which could explain why, in this cohort, STING appears to be involved in lipid accumulation in the SS stage, a condition that does not manifest as inflammation. Therefore, although STING has been previously reported to be a mediator of inflammation in the liver [25], we were unable to confirm this role in the current study.

Given these results, which seem to suggest that STING is mostly involved in the lipid accumulation process in the liver, hepatic STING expression was evaluated according to the degree of steatosis of the samples. The results showed increased expression of STING in cases of mild and moderate degrees of steatosis compared with the absence of steatosis. This finding is logical since in these patients, STING seems to inhibit lipid homeostasis, which triggers lipogenic and proinflammatory cascades [25,42]. Indeed, this function might be relevant in the first steps of lipid accumulation in the SS stage. However, these findings are preliminary and need to be further studied and validated.

To further elucidate the involvement of STING in NAFLD, correlations with some parameters related to this disorder, such as inflammatory factors, lipid metabolism-related genes, liver enzymes, Toll-like receptors, and microbiota-derived bile acids, were analyzed. Positive correlations were found between STING mRNA expression in the liver and GGT and ALP levels. Although the differences in GGT levels among the three groups (NL, SS, and NASH) were not significant, a significant increase was observed in the GGT concentration in these NAFLD patients. This result is in concordance with the study by Neuman et al. [43]. In fact, high levels of GGT are considered a consequence of NAFLD [44]. Moreover, a study by Zhou et al. reported that elevated levels of ALP might be considered a relevant predictor of NAFLD progression in a cohort composed of women of reproductive age [45]. In this case, patients of the present cohort who were in the SS stage showed increased levels of this enzyme compared with patients with NL; patients with NL also showed a decrease in NASH compared with SS subjects. This pattern is in line with what has been reported in terms of hepatic STING mRNA and protein expression. Thus, given that STING seems to be involved in NAFLD pathogenesis and that GGT and ALP are two of the main liver enzymes whose expression is enhanced in NAFLD, these correlations are reasonable and in accordance with previous results. Unfortunately, we did not find significant correlations between STING mRNA expression in liver and ALT or AST circulating levels, which also showed nonsignificant differences between groups. These findings regarding to hepatic transaminases levels can be explained, since it has been previously reported that liver transaminases levels do not always correspond exactly with the hepatic histological stage of NAFLD [46].

Likewise, another positive association between STING and TLR9 hepatic mRNA abundance was observed. This finding can be explained in that TLR9 is linked to the recognition of microbiota-derived metabolites that result from intestinal dysbiosis and is capable of recognizing mtDNA from injured hepatocytes, such as that which results from the secretion of the proinflammatory cytokine IL-1β, which has a crucial role in NASH progression [47]. On the contrary, STING has been reported to be related to NAFLD pathogenesis and to inflammatory processes involved in NASH [26]. Although they do not stimulate the production of the same cytokines, both STING and TLR9 contribute to the occurrence of inflammation in NAFLD [26,47]. However, the current results show that hepatic STING expression was not significantly different between patients with hepatic inflammation or ballooning compared with subjects without these parameters. This finding reinforces the previous proposed hypothesis that in these patients, STING was not strongly implicated in proinflammatory processes but could be implicated in metabolic pathways. Nevertheless, no significant correlation was found between hepatic STING expression and the expression of any lipid metabolism-related genes. For these reasons, the association between TLR9 and STING should be further investigated.

Additionally, positive associations were found between hepatic STING expression and several microbiota-derived bile acids, such as GCDCA, GCA, GDCA, GUDCA, TCA, and TUDCA. Evidence already shows that STING can modify intestinal bacteria [38], which in turn leads to modifications in bile acid secretion. In fact, compared with wild-type mice, different gut bacteria were found in mice in which STING expression was suppressed, although all mice were fed the same high-fat diet (HFD). Despite being fed an HFD, the authors revealed that mice in which the STING gene was knocked out had intestinal microbiota that was richer and more evenly distributed, which is associated with a better hepatic condition [38]. In the present study, the potential association between STING function and bile acid metabolism, both of which are related to NAFLD pathogenesis through the gut–liver axis, was reinforced [32,48].

Curiously, although STING has been shown to be related to proinflammatory processes of NAFLD in previous reports [32,41,47], in agreement with aforementioned results, only nonsignificant correlations were observed between hepatic STING expression and circulating proinflammatory markers, such as cytokines [49]. In our patients, hepatic STING seems to be primarily involved in lipid metabolism rather than in inflammatory processes characteristic of NASH, both in the mRNA and protein analyses. It is important to note that the cohort of patients in this study included only women, all of whom presented with MO. The condition of obesity may interfere with the evaluation of the inflammatory processes of STING, since these patients exhibited a low-grade chronic inflammation pattern [50]. In addition, these patients were under a very low-calorie diet three weeks prior to surgery [51]. This strict diet could have affected the determinations made. Additionally, these results cannot be extrapolated to other populations, such as lean subjects, men, or free-diet participants. In any case, the current study proposes for the first time a possible key implication of STING in hepatic lipid metabolism in patients with NAFLD-associated obesity.

## 5. Conclusions

In summary, the present study suggested that in liver tissue, STING appears to be involved in the development of NAFLD. As such, hepatic STING mRNA and protein expression, which was reported to be involved in lipid metabolism, is increased in the SS stage. However, the proinflammatory action of STING could be masked by other proinflammatory mediators in patients with MO. In addition, hepatic STING mRNA expression was positively correlated with several characteristic NAFLD parameters, such as liver enzyme levels, hepatic TLR9 mRNA expression, and microbiota-derived bile acid levels. The results of this study reinforced the possible key role of STING in NAFLD pathogenesis. Nevertheless, further studies are necessary to validate these findings.

## Figures and Tables

**Figure 1 metabolites-13-00496-f001:**
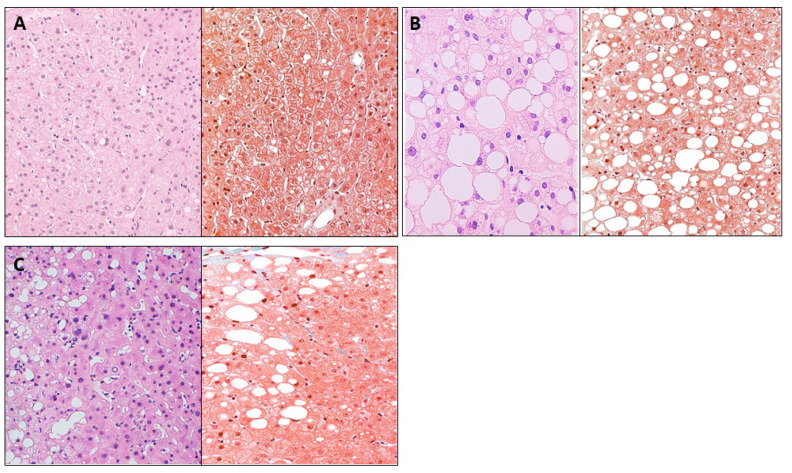
Histological stains with hematoxylin and eosin and Masson’s trichrome stains of women with MO, (**A**) NL, subject with SS (**B**), and patients with NASH (**C**). NL samples show a physiological phenotype. SS presents fat accumulation in more than 5% of the tissue, whereas in NASH, in addition to excessive fat accumulation, inflammatory cells (dark blue) and scant perisinusoidal fibrosis are detected. ×400. Normal liver, NL; Simple steatosis; SS; nonalcoholic steatohepatitis, NASH.

**Figure 2 metabolites-13-00496-f002:**
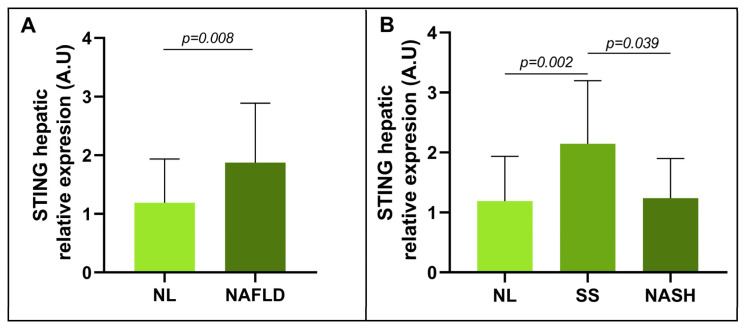
Differential relative mRNA abundance of STING in liver samples from women with MO (**A**) classified as NL and NAFLD and (**B**) classified as NL, SS, and NASH. STING, stimulator of interferon genes; NL, normal liver; SS, simple steatosis; NASH, nonalcoholic steatohepatitis; A.U, arbitrary units. Mann–Whitney test was used to calculate the difference between groups considering *p* < 0.05 statistically significant.

**Figure 3 metabolites-13-00496-f003:**
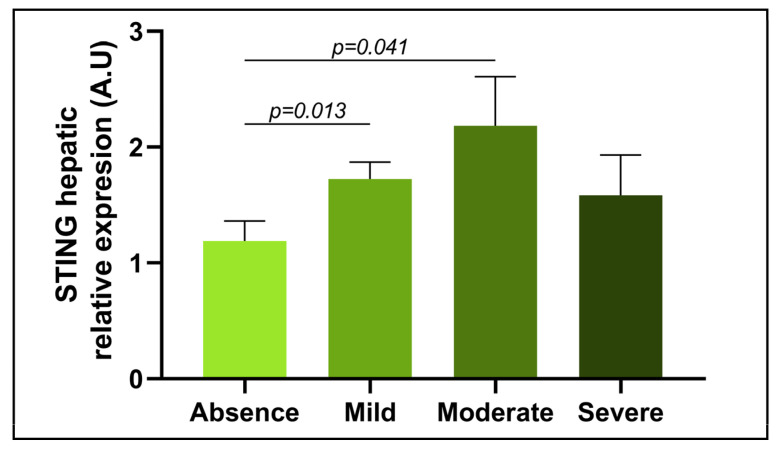
Differential relative mRNA abundance of STING in liver samples from women with MO classified according to different grades of steatosis into absence, mild, moderate, and severe. STING, stimulator of interferons genes; A.U, arbitrary units. Mann–Whitney test was used to calculate the difference between groups considering *p* < 0.05 as statistically significant.

**Figure 4 metabolites-13-00496-f004:**
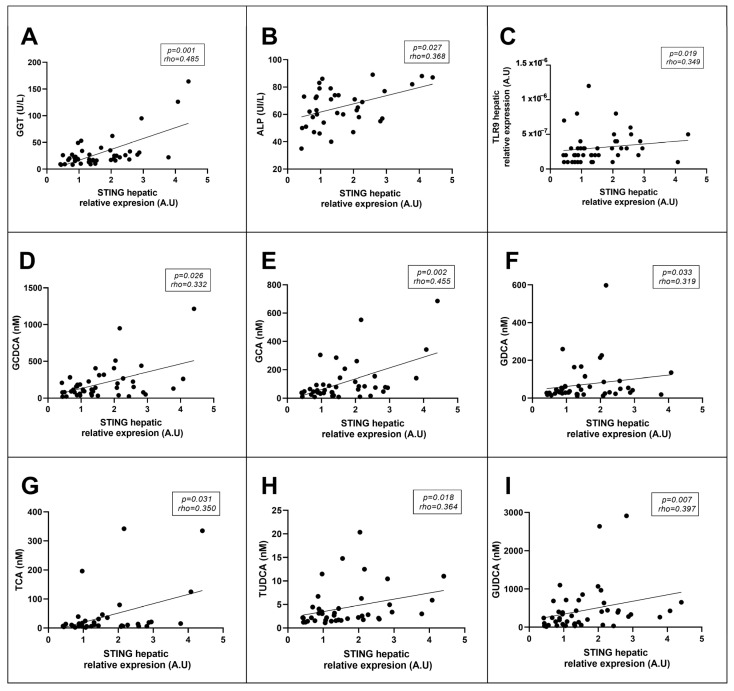
Spearman’s method was used in order to identify significant correlations between STING hepatic mRNA expression and (**A**) GGT, (**B**) ALP, (**C**) TLR9, (**D**) GCDCA, (**E**) GCA, (**F**) GDCA, (**G**) TCA, (**H**) TUDCA, and (**I**) GUDCA. STING, stimulator of interferon genes; GGT, gama-glutamyl transferase; ALP, alkaline phosphatase; TLR9, toll-like receptor 9; GCDCA, glycochenodeoxycholic acid; GCA, glycocholic acid; GDCA, glycodeoxycholic acid; TCA, taurocholic acid; TUDCA, tauroursodeoxycholic acid; GUDCA glycoursodeoxycholic acid; A.U arbitrary units. *p* < 0.05 was considered statistically significant. Spearman correlation coefficient (rho).

**Figure 5 metabolites-13-00496-f005:**
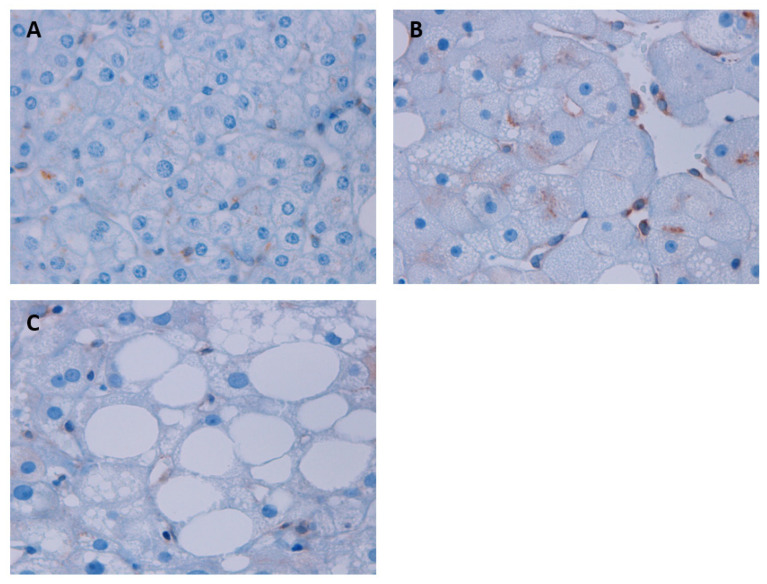
Representative images of the hepatic biopsies where the IHC analysis of STING was carried out (×400) in (**A**) NL, (**B**) SS, and (**C**) NASH groups. STING protein expression was identified as brown dots, being observed mainly in nonparenchymal cells.

**Table 1 metabolites-13-00496-t001:** Biochemical and anthropometric variables of women in the studied cohort.

Variables	NL (n = 27)	SS (n = 26)	NASH (n = 16)
Weight (kg)	117 (107–131)	114 (108.98–128.6)	110.5 (104.33–120.75)
BMI (kg/m^2^)	43.50 (40.89–46.88)	44.35 (40.87–46.8)	44.19 (40.69–45.81)
SBP (mmHg)	120 (100–132.5)	117.5 (108.5–127)	115 (102–127)
DBP (mmHg)	63 (57.5–73)	62 (59.5–73.75)	64 (55–70)
Glucose (mg/dL)	85.04 (76.03–93.14)	93.14 (87.2–107.02) *	91.52 (82.33–101.97)
Insulin (mUI/L)	9.57 (5.55–16.82)	10.17 (7.23–13.93)	7.19 (5.14–26.02)
HbA1c (%)	5.5 (5.3–5.7)	5.55 (5.3–5.95)	5.55 (5.15–6.13)
HOMA1	2.05 (1.03–3.45)	2.52 (1.38–3.68)	1.63 (1.26–4.23)
Cholesterol (mg/dL)	170 (148.25–209.5)	171.15 (136.25–194.25)	183.9 (152.75–229.5)
HDL-c (mg/dL)	40.6 (32.05–48.5)	43.5 (33.75–47)	37.8 (33.5–48.5)
LDL-c (mg/dL)	107.9 (86–134.2)	104.1 (77.20–126.25)	94 (79.3–128.03)
TG (mg/dL)	106.5 (94–136)	117.5 (82.25–172.5)	153 (116.5–256.5) *
AST (UI/L)	20 (15.5–36.5)	23 (17–35)	27 (17.25–43.5)
ALT (UI/L)	22.5 (16–37.5)	31 (22–35.25)	32 (16.25–41)
GGT (UI/L)	18 (15.25–26.25)	21 (16–32.25)	25.5 (18–28.75)
ALP (UI/L)	58.5 (49.25–71.25)	74 (64–86.25) *	63 (55–74.5) ^$^

NL, normal liver; SS, simple steatosis; NASH, nonalcoholic steatohepatitis; BMI, body mass index; SBP, systolic blood pressure; DBP, diastolic blood pressure; HOMA1-IR, homeostatic model assessment method-insulin resistance; HbA1c, glycosylated hemoglobin; TG, triglycerides; HDL-C, high-density lipoprotein cholesterol; LDL-C, low-density lipoprotein cholesterol; AST, aspartate aminotransferase; ALT, alanine aminotransferase; GGT, gamma-glutamyl transferase; ALP, alkaline phosphatase. Data are expressed as the median (interquartile range). * Significant differences vs. NL group (*p* < 0.05). ^$^ Significant differences vs. SS group (*p* < 0.05).

## Data Availability

Data is unavailable due to privacy.

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
