# Peer review of "Expression of STING in Women with Morbid Obesity and Nonalcoholic Fatty Liver Disease"

_metabolites, 2023, doi:10.3390/metabo13040496_

Round 1

Reviewer 1 Report (Previous Reviewer 1)

The authors have addressed some of my concerns in the revised manuscript. Still, I have the comments shown below that need to be clarified. 1: Figure 1 should be described in the "Results" section.  It is recommended to provide high-quality images to better demonstrate the level of fibrosis, as Masson's trichrome staining results shown in Figure 1 are of poor quality. 2: The hepatic morphology should be consistent between Figure 5 and Figure 1, especially regarding steatosis.

Author Response

Reviewer 2 Report (Previous Reviewer 2)

The authors have significantly improved the manuscript by adding IHC staining. I don't have further comments

Author Response

We appreciate so much the valuable suggestions you made to improve the quality of our manuscript. Thank you.

Round 2

Reviewer 1 Report (Previous Reviewer 1)

No more comments

This manuscript is a resubmission of an earlier submission. The following is a list of the peer review reports and author responses from that submission.

Round 1

Reviewer 1 Report

In the manuscript entitled “Expression of STING in Women with Morbid Obesity and Nonalcoholic Fatty liver Disease,” the author analyzed the hepatic STING mRNA abundance in women with morbid obesity and found that hepatic STING mRNA expression increases with the occurrence of NAFLD, specifically in the simple steatosis stage and positive correlation of STING expression with gamma-glutamyl transferase and alkaline phosphatase, Toll-like receptor 9 expression and some circulating microbiota-derived bile acids. The results from this study are interesting, however, the paper suffers from some problems, which detract from the overall quality. I have some comments shown below to help improve the clarity of this manuscript.  

  1. As an article paper, it would be better to remove Figure 1 from the manuscript, which has already been described in the introduction. 
  2. H&E and Masson’s trichome staining results of control, NAFLD, SS, and NASH groups should be included in the manuscript. 
  3. The primer sequences used for RT-PCR should be provided in the “Materials and Methods” 
  4. In Table 1, there are no differences in the levels of AST and ALT between NL and NASH groups, it would be better to provide an explanation. 
  5. For Figure 2 and Figure 3, STING protein levels should be included to validate the change in mRNA levels. 
  6. For Figure 2 and Figure 3, Pearson Correlation analysis should be used. 
  7. “Data not shown” appears in Line 277, it would be better to show these results. 

Reviewer 2 Report

In this work, Bertran and colleagues investigate the possible role of STING in NAFLD by evaluating its hepatic mRNA expression and doing several correlation analysis. In my opinion the manuscript needs important experiments to validate these findings. For instance, the authors should determine:

1) The protein levels of STING in liver tissue either by WB or IHC.

2) The circulating levels of STING (as the propose a direct role in gut microbiota).

In addition, the authors propose that a role of STING on the microbiota, but they merely perform association studies with circulating bile acids. Therefore, correlation between STING (hepatic mRNA and protein, and circulating levels) gut microbiota composition should directly analyzed instead of using circulating metabolites.

Round 2

Reviewer 1 Report

The authors have addressed some of my concerns in the revised manuscript. Still, I have the comments shown below that need to be clarified.

  1. The Masson’s trichome staining was described in the “Materials and Methods” and “Result”. The images should be included in the manuscript. 
  2. The protein levels are needed to validate the changes in STING mRNA levels. IHC staining with the liver sections should be included if liver biopsies are not available. 

Reviewer 2 Report

I still consider that the authors should measure STING protein in livers. If liver biopsies are not available, they should at least try to get paraffin-embedded tissue sections to perform IHC in a representative group of patients.

Re microbiota, could the authors include a brief explanation of how STING could affect gut microbiota?